# The Two-Component System CpxRA Affects Antibiotic Susceptibility and Biofilm Formation in Avian Pathogenic *Escherichia coli*

**DOI:** 10.3390/ani13030383

**Published:** 2023-01-23

**Authors:** Kai Ma, Hui Wang, Zhenfei Lv, Yutong Hu, Hongli Wang, Fang Shu, Chengfeng Zhu, Ting Xue

**Affiliations:** School of Life Sciences, Anhui Agricultural University, Hefei 230036, China

**Keywords:** avian pathogenic *Escherichia coli*, CpxRA two-component system, antibiotic susceptibility, biofilms

## Abstract

**Simple Summary:**

Avian pathogenic *Escherichia coli* (APEC) is the main pathogen of colibacillosis and impedes the development of the poultry industry. It is of great significance to study the mechanism of drug resistance in APEC. In this study, we investigated the role of the two-component system CpxRA in APEC. The results showed that *cpxRA* was involved in bacterial biofilm formation, antibiotic susceptibility, and transcription levels of efflux pump *emrKY*. Our data suggest that the two-component system *cpxRA* is indeed involved in APEC resistance.

**Abstract:**

Avian pathogenic *Escherichia coli* (APEC) is one of the common extraintestinal infectious disease pathogens in chickens, geese, and other birds. It can cause a variety of infections, and even the death of poultry, causing enormous economic losses. However, the misuse and abuse of antibiotics in the poultry industry have led to the development of drug resistance in the gut microbes, posing a challenge for the treatment of APEC infections. It has been reported that the CpxRA two-component system has an effect on bacterial drug resistance, but the specific regulatory mechanism remains unclear. In this study, the regulatory mechanism of CpxRA on APEC biofilm formation and EmrKY efflux pump was investigated. The *cpxRA* knockout strain of *E. coli* APEC40 was constructed, and the molecular regulatory mechanism of CpxR on biofilms and efflux pump-coding genes were identified by biofilm formation assays, drug susceptibility test, real-time reverse transcription quantitative PCR, and electrophoretic mobility shift assay (EMSA). The results indicated that CpxR can directly bind to the promoter region of *emrKY* and negatively regulate the sensitivity of bacteria to ofloxacin and erythromycin. These results confirm the important regulatory role of the CpxRA two-component system under antibiotic stress in APEC.

## 1. Introduction

Avian pathogenic *Escherichia coli* (APEC), an extraintestinal pathogenic *Escherichia coli* (ExPEC), is one of the frequent causative agents causing extraintestinal infectious diseases of avians such as chickens and geese. It can cause local and systemic infections such as pericarditis, salpingitis, perihepatitis, septicemia, spondylitis, and other extraintestinal infectious diseases in avians, collectively referred to as avian colibacillosis [1,2,3,4]. Colibacillosis can lead to decreased weight and egg production of poultry and even result in slaughter carcass necrosis [5]. The high morbidity and mortality rates caused by colibacillosis cause millions of dollars in economic losses to the poultry industry every year, seriously hindering the development of the poultry industry. Although the poultry breeding system has undergone improvement, colibacillosis is still a serious problem in the poultry industry worldwide [6,7].

At present, antibiotics are still the main means to reduce the incidence rate and mortality of avian colibacillosis [8,9]. However, the misuse and abuse of antibiotics have led to multidrug resistance in APEC, which increases the difficulty of treatment and poses challenges to the control of APEC infections [10]. APEC has some similar characteristics to uropathogenic *Escherichia coli* (UPEC) and neonatal meningitis *Escherichia coli* (NMEC) [11,12]. They can be transmitted to humans through foodborne cross-infection by handling live poultry or eating undercooked poultry products, which indicates the risk of zoonosis [13,14]. Therefore, APEC may be a potential danger to human health. 

In general, bacterial resistance involves several mechanisms, such as modification of antimicrobial targets to reduce drug affinity; modifications of the antimicrobial molecule; change in membrane permeability to reduce drug uptake; activation of the efflux pump to pump drugs out of the cell; and resistance due to global cell-adaptive processes [15,16]. The expression of efflux pump genes is a common mechanism of bacterial multidrug resistance [17]. According to the energy dependence and structure, five major efflux pump families have been identified, which are the ATP-binding cassette (ABC) superfamily, the RND superfamily, the major facilitator superfamily (MFS), the small multidrug resistance family (SMR), and the multiantimicrobial extrusion (MATE) family [18]. The *E. coli* K-12 genome contains 36 known or putative drug efflux pump-coding genes, with highly conserved sequences, which can extrude one or more drugs from bacterial cells, causing bacterial multidrug resistance [19,20].

Biofilm is a dynamically changing bacterial aggregate with a complex structure, which is an important and special growth mode of bacteria. It can protect bacterial cells from hostile environments and help the bacteria within them disperse to new niches [21]. The biofilm formation process includes five stages: early attachment, reproduction, efflux, maturation, and diffusion. Bacteria in biofilm are difficult to remove and kill, resulting in persistent infection of the host [22,23].

A two-component system (TCS) is a signal transduction system by which bacteria sense external signals and make corresponding changes to adapt to environmental changes [24]. CpxRA is a TCS, which consists of a sensor histidine kinase CpxA located in the bacterial inner membrane and a response regulator CpxR in the cytoplasm [25]. CpxRA is involved in the viability and pathogenicity of bacteria, including virulence, antibiotic resistance, biofilm formation, and oxidative stress resistance [26,27,28]. However, the mechanism of resistance to APEC regulated by the CpxRA TCS remains unclear.

This study aimed to elucidate the effect of CpxRA on antibiotic susceptibility and biofilm formation properties of the strain APEC40. In the present study, we constructed a *cpxRA* deletion mutant strain using the λ Red recombinase system [29]. The biofilm formation ability and the sensitivity to erythromycin and ofloxacin of the *cpxRA* mutant strain and WT strains were compared. In addition, the regulation mechanism of CpxR on APEC40 multidrug efflux pump EmrKY was investigated and studied by reverse transcription qPCR (RT-qPCR) and electrophoretic mobility shift assay (EMSA). Therefore, this study helps to understand the regulatory mechanism of APEC under antibiotic pressure, and provides a new experimental basis for the treatment of *E. coli* infection.

## 2. Materials and Methods

### 2.1. Bacterial Strains, Plasmids, and Culture Conditions

The bacterial strains and plasmids used in this study are listed in Table 1. *E. coli* strains were routinely grown aerobically in lysogeny broth (LB) medium at 37 °C with constant shaking at 200 rpm or maintained on LB agar plates, except when indicated. All cultures for pKD46 or pCP20 temperature-sensitive plasmid maintenance were incubated at 30 °C. Cell growth was monitored by measuring the absorbance at 600 nm. Ampicillin (Amp), kanamycin (Kan), and chloramphenicol (Cm) were used for plasmid selection and maintenance, at 100, 50, and 30 μg/mL, respectively.

### 2.2. Genetic and Molecular Biology Techniques

Molecular manipulation was performed according to a standard protocol for Gram-negative bacteria. *E. coli* APEC40 (WT) genomic DNA and plasmid DNA were extracted using a genome extraction kit (Sangon Biotech, Shanghai, China) and plasmid extraction kit (Transgen, Beijing, China), respectively. PCR amplification was conducted by using Taq or PrimeSTAR^®^ Max DNA Polymerase (Takara Bio Inc., Dalian, China). A gel purification kit was used to purify the PCR products and DNA restriction fragments (Transgen, Beijing, China). DNA restriction endonuclease (Thermo Fisher Scientific, Waltham, MA, USA) digestion and homologous recombination (Vazyme, Nanjing, China) were conducted using standard methods. The primer sequences used are listed in Table 2.

### 2.3. Construction of the CpxRA-Deficient Mutant and Complemented Strains

Based on the λ Red recombinase system, the *cpxRA* operon gene all deletion mutant of APEC40 was constructed by homologous recombination [29]. Briefly, this involved using primers APEC40-*cpxRA*-f and APEC40-*cpxRA*-r to amplify the chloramphenicol resistance cassette (*cat*) from the pKD3 plasmid to replace the target gene in WT, and the fragment was electroporated into APEC40 competent cells containing λ Red recombinase expressed from the pKD46 plasmid. After electroporation, shocked cells were added immediately to 900 μL LB without antibiotics, and incubated at 37 °C for 1 h. Then, the cultures were spread on LB plates with 30 μg/mL Cm for screening the mutants. The *cpxRA* mutant strains were screened by PCR amplification by primers Check-*cpxRA*-in-f/r, Check-*cpxRA*-out-f/r. The resulting mutants were inoculated into LB containing 30 μg/mL chloramphenicol and incubated overnight at 42 °C with aeration to clear the plasmid pKD46. Immediately thereafter, the plasmid pCP20 was transferred into the mutant strain to cure the *cat*. The resulting Cm-sensitive mutant strain was confirmed by PCR amplification with the primers Check-*cpxAR*-out-f and Check-*cpxRA*-out-r, and the mutant strain was further confirmed by DNA sequencing. The mutant strain was named APEC40/*ΔcpxRA* (XM1).

For functional complementation of the *cpxRA* mutant strain, the open reading frame (ORF) of the *cpxRA* gene, and the promoter were amplified from the APEC40 strain using the primers pC*cpxRA*-kpnI-f and pC*cpxRA*-hindIII-r, subcloned into the low-copy plasmid pSTV28 (Takara, Dalian, China) by enzymatic digestion and DNA ligase technology, and then the recombinant plasmid pC*cpxRA* was electroporated into the mutant strain XM1 to obtain the complemented strain XM1/pC*cpxRA*. As a control, the wild-type strain (WT) and the mutant strain XM1 were also transferred with the empty pSTV28 plasmid to obtain WT/pSTV28 and XM1/pSTV28, respectively.

### 2.4. Bacterial Growth Curves

The overnight cultures of WT/pSTV28, XM1/pSTV28, and XM1/pC*cpxRA* were diluted in 50 mL fresh liquid LB medium with 30 μg/mL chloramphenicol to a certain concentration (OD_600_ = 0.03). The diluted bacteria were grown at 37 °C for 24 h with shaking. An ultraviolet visible spectrophotometer (Thermo Scientific, Pittsburgh, PA, USA) was used to detect the cell density at 600 nm of ultraviolet light every 2 h. Colony-forming unit (CFU) counting experiments were conducted at 2 h, 4 h, and 10 h. The experiment was repeated independently 3 times.

### 2.5. Biofilm Formation Assays

The ability of biofilm formation was determined by growing the biofilm in sterile polystyrene tubes according to a previous and modified method [30]. Briefly, the overnight cultures of WT/pSTV28, XM1/pSTV28, and XM1/pC*cpxRA* were diluted in a ratio of 1:100 in fresh LB containing 30 μg/mL Cm and 3 mL aliquots were transferred to polystyrene tubes. After incubation for 20 h at 37 °C, the medium was discarded and the tubes were rinsed 3 times with sterile phosphate-buffered saline (PBS) and air-dried. The biofilm formation cells were fixed with 100% methanol for 5 min, washed, and stained for 20 min with 0.1% (*w*/*v*) crystal violet (CV) (Sangon, Shanghai, China). After washing and drying, the adsorbed dye was dissolved with 33% glacial acetic acid (Sangon, Shanghai, China). Subsequently, the absorbance was read at 492 nm by a MicroELISA Autoreader (Thermo Scientific, Pittsburgh, PA, USA) in single-wavelength mode to quantitatively detect the biofilm. The experiment was repeated independently 3 times.

### 2.6. Antibiotic Susceptibility Tests

According to the Clinical and Laboratory Standard Institute (CLSI), the changes in antimicrobial susceptibility of the *cpxRA* mutant XM1/pSTV28 and the complementation strain XM1/pC*cpxRA* and strain WT/pSTV28 were examined by using Mueller–Hinton (MH) broth dilution antimicrobial susceptibility tests. The lowest concentration of antimicrobials that completely inhibited growth was identified as the minimal inhibitory concentration (MIC). The experiment was repeated 3 times. The antimicrobial agents tested were: ofloxacin (fluoroquinolone, Sangon) at 20 mg/mL, erythromycin (macrolide, Sangon) at 30 mg/mL.

### 2.7. Antibacterial Activity Assays

Referring to previous methods [31], the antibiotic susceptibility of WT/pSTV28, XM1/pSTV28, and XM1/pC*cpxRA* was determined by antibacterial activity assays. Overnight cultures of WT/pSTV28, XM1/pSTV28, and XM1/pC*cpxRA* were diluted in fresh LB medium containing 30 μg/mL Cm to a certain concentration (OD_600_ = 0.03), respectively, at 37 °C for 2 h with shaking. After incubation, different antibiotics (final concentration 1/2MIC) were added to the culture tubes and incubated at 37 °C for 3 h. Cultures were then serially transferred 10-fold (0.1 mL) to 8 EP tubes containing 0.9 mL MH for serial dilution of cultures. Colony counts (CFU/mL) were performed by the microdilution plating method in which three appropriate dilutions (0.1 mL) were dropped onto LB agar plates and colonies were counted after incubation at 37 °C. The survival rates of WT/pSTV28 were designated as 100%. The experiments were repeated 3 times.

### 2.8. Total RNA Isolation, cDNA Generation, and Real-Time PCR Processing

The overnight cultures of WT/pSTV28, XM1/pSTV28, and XM1/pC*cpxRA* were inoculated into 100 mL fresh LB at 1:100 and incubated at 37 °C with shaking The bacteria were collected for total RNA extraction when they reached the exponential phase. The Spin Column Bacteria Total RNA Purification Kit (Sangon) was used to extract total RNA from cells after collecting the cells by centrifugation and resuspending them in RNase-free water in advance. Reverse transcription was performed using the EasyScript One-Step gDNA Removal and cDNA Synthesis SuperMix kit (Transgen), according to the manufacturer’s instructions. RT-qPCR was performed with RT primers following the instructions of the TransStart Tip Green qPCR SuperMix kit (Transgen) on the CFX96 Real-Time System (BioRad). With the 16S cDNA gene as a housekeeping gene, the quantity of the target genes was normalized [32]. All of the RT-qPCR assays were repeated at least 3 times.

### 2.9. His6-CpxR Protein Purification

The *cpxR* gene was amplified using primers pET-*cpxR*-f and pET-*cpxR*-r from the genomic DNA of the WT strain and then cloned into the prokaryotic expression plasmid pET-28a. The recombinant plasmid pET-*cpxR* was transferred into the BL21(DE3) strain by a chemical transformation method and grown in LB containing 50 μg/L kanamycin at 37 °C with shaking to OD_600_ of 0.5–0.8, then isopropyl β-D-thiogalactopyranoside (IPTG) was added to the final concentration of 0.5 mM and the mixture were incubated overnight at 16 °C for 12 h. His6-CpxR was purified by Ni-NTA resin affinity chromatography [32]. The concentration of purified protein was measured using the BCA Protein Assay Kit (Beyotime, Shanghai, China) and the protein was stored in 10% glycerol at −80 °C until use.

### 2.10. Electrophoretic Mobility Shift Assays (EMSAs)

The promoters of *emrKY* genes were amplified by p-*emrKY*-biotin-f/p-*emrKY*-r using the chromosomal DNA of APEC40 as a template. Increasing amounts of His6-CpxR protein were added to biotin-labeled promoter DNA, and binding reactions were performed with 4 µL 5 × binding buffer: 50 mM Tris–HCl (pH 7.5), 100 mM NaCl, 3 mM magnesium acetate, 0.1 mM EDTA, 0.1 mM dithiothreitol. Reaction mixtures were incubated for 30 min at room temperature. When required, the unlabeled DNA fragments were added as competitive probes. After incubation, bromophenol blue loading buffer was added to the mixture, followed by electrophoresis in a 4% native polyacrylamide gel in a 0.5 × Tris–borate EDTA buffer. DNA bands were detected and analyzed according to the manufacturer’s chemiluminescent EMSA kit instructions (Beyotime, Shanghai, China).

### 2.11. Statistical Analyses

All data were statistically analyzed using GraphPad Prism 8.0 (GraphPad Software Inc., GraphPad Prism 8.0.2, San Diego, CA, USA, 2018). The test results were shown as the mean ± SD. Comparison between two groups was conducted by using a *t*-test. *P* ≤ 0.05 indicates a significant difference.

## 3. Results

### 3.1. Identification of cpxRA Mutant and Complementary Strains of APEC40

The *cpxRA* deletion mutant strain was obtained by knockout of the *cpxRA* gene through homologous recombination. The mutant and complementary strains of cpxRA were identified by PCR (Figure 1A). The products of lanes 1–3 were 2465 bp (amplified from the WT strain WT/pSTV28), 596 bp (amplified from the mutant XM1/pSTV28), and 596 bp (the complementary strain XM1/pCcpxRA), respectively. The plasmid pSTV28 and the complementary plasmid pCcpxRA were confirmed by PCR. The products of lanes 4–6 were 150 bp (amplified from strain WT/pSTV28 with primers M13-f/M13-r), 150 bp (amplified from strain XM1/pSTV28 with primers M13-f/M13-r), and 2393 bp (amplified from strain XM1/pCcpxRA).

### 3.2. cpxRA Deletion Did Not Affect Strain Growth

To confirm whether there was an effect of *cpxRA* gene deletion on the growth trend of the WT strain, growth curve and CFU counting experiments were performed. The growth trend of each strain was measured under the same culture conditions. The results showed that the growth trends of WT/pSTV28, XM1/pSTV28, and XM1/pC*cpxRA* strains in LB medium containing 15 μg/mL Cm were similar (Figure 1B), and in the lag, exponential, and stationary phase of the growth curve, there was no significant difference in CFUs (Figure 1C–E), indicating that the absence of cpxRA did not affect the growth of the WT strain.

### 3.3. Deletion of cpxRA Inhibited Biofilm Formation

The biofilm of a pathogen is important in host infection. Therefore, we performed a CV staining assay to measure solid-surface-associated biofilm formation. Biofilm formation of WT/pSTV28, XM1/pSTV28, and XM1/pC*cpxRA* in polystyrene tubes was observed after 20 h of incubation in LB containing 30 μg/mL Cm. As shown in Figure 2, the formation of solid-surface-conjugated biofilms of XM1/pSTV28 in polystyrene tubes was reduced by 1.58-fold compared with WT, and the biofilm formation ability was restored in XM1/pC*cpxRA*.

### 3.4. Deletion of the cpxRA Gene Decreased Bacterial Antibiotic Susceptibility

The MIC of experimental bacteria for the two antibiotics was determined according to the CLSI standards. The MICs are shown in Table 3. The results indicated that the MICs of XM1/pSTV28 and XM1/pC*cpxRA* for the two antibiotics were similar to that of WT/pSTV28. 

To further determine the effect of *cpxRA* on the drug sensitivity of the APEC40 strain, bacterial cultures were grown in LB containing 30 μg/mL chloramphenicol and then treated with different antibiotics. Finally, the viable count of WT/pSTV28, XM1/pSTV28, and XM1/pC*cpxRA* was determined using the CFU counting method. In the presence of ofloxacin and erythromycin, the viable count of XM1/pSTV28 increased approximately 1.76-fold and 1.48-fold (*p* < 0.05), respectively, when compared with WT/pSTV28. The viable count of XM1/pC*cpxRA* fell to that of WT/pSTV28 (Figure 3). These data illustrated that the deletion of the *cpxRA* gene significantly reduced the sensitivity of APEC40 to the above two antibiotics.

### 3.5. CpxRA Down-Regulated the Transcription Level of emrKY

To investigate how CpxR regulates bacterial susceptibility to antibiotics, we performed RT-qPCR experiments to detect the transcription levels of genes encoding multidrug efflux pumps, including *emrK*, *emrY*. The results showed that the transcription levels of *emrK* and *emrY* in XM1/pSTV28 increased by 4.12-fold and 3.07-fold, respectively, compared with WT/pSTV28 (Figure 4). The transcription levels of these two genes were restored in XM1/pC*cpxRA*. These data suggested that the *cpxRA* gene decreases the antibiotic sensitivity of APEC40 by regulating the genes related to efflux pumps.

### 3.6. CpxR Binding to EmrKY Promoters

According to the above results, we hypothesized that CpxR regulates the transcription level of the *emrKY* by directly binding to the *emrKY* gene promoter. EMSA experiments were performed to strengthen this speculation. The positive control showed that CpxR was able to bind to its promoter [33], and significant protein–DNA complex band shifts were detected at CpxR concentrations of 1, 2, and 4 μM, and the intensity of the shifted bands increased with enhanced CpxR concentration (Figure 5). The results confirmed that CpxR can specifically bind the promoter of *emrKY*, indicating that CpxR can directly regulate the transcription of the EmrKY efflux pump.

## 4. Discussion

All organisms have ways of resisting adverse environments, and the environmental stress tolerance of microorganisms in particular is more complex and diverse, making it a daunting challenge to control bacterial infection in humans and livestock and poultry. Therefore, there is an urgent need to explore the survival mechanism of bacteria in the face of adverse conditions and find potential drug targets. Biofilm is an important feature of *E. coli* that protects cells from adverse environmental conditions [34]. In this study, the regulation of biofilm-related properties of the APEC40 strain by the CpxRA TCS was investigated. Our results confirmed that the allelic mutant *ΔcpxRA* leads to a reduced biofilm formation. In addition, the efflux pump is also an important factor in bacterial resistance to antibiotics to ensure survival [17]. The results showed that *cpxRA* gene knockout promoted the survival of *E. coli* under antibiotics erythromycin and ofloxacin compared to the WT strain. The underlying mechanism was that CpxR bound directly to the promoter region of the efflux pump gene *emrKY*, and negatively regulated the transcription level.

The biofilm formation of *E. coli* helps a variety of infections occur, making them difficult to eradicate. The colonization and formation of mature biological envelopes are finely regulated [35]. In a previous study, OmpA inhibited cellulose production through the CpxRA stress response system, and the decrease in cellulose increased biofilm formation on hydrophobic surfaces in *E. coli* [36]. In *Actinobacillus pleuropneumoniae*, CpxA/CpxR has been shown to contribute to biofilm formation [26].

However, the effect of CpxR on the ability of biofilm formation in APEC is rarely reported. Our results indicated that the function of *cpxRA* in APEC40 was similar to what has been reported. The formation of solid-surface-bound biofilm in polystyrene tubes in XM1/pSTV28 was significantly reduced. Therefore, we consider that the Cpx signaling pathway plays an important regulatory role in the biofilm formation in *E. coli* APEC40, but the specific regulatory mechanism needs to be further studied.

In *E. coli*, the two-component system CpxRA can regulate some physiological processes and is an important regulatory system. More recently, the CpxRA system has been reported to respond to antibiotics by altering the membrane integrity and increasing antimicrobial resistance [37]. In addition, the Cpx signaling pathway responds to stress from misfolded proteins in the inner membrane [38] as well as to a lipoprotein, NlpE, located on the outer membrane [39]. However, much is still unknown about what target genes CpxR regulates and its role in resistance to environmental stress. In particular, it is unknown whether CpxR modulates drug resistance in APEC. Gram-negative bacteria have many efflux pumps that pump structurally diverse molecules, including antibiotics. One of the reasons bacteria can survive higher concentrations of antibiotics is that this efflux lowers the intracellular concentration of antibiotics [40]. In *Salmonella* Typhimurium, the transporter protein IceT causes citrate efflux, reduces intracellular iron content, and reduces sensitivity to oxidative stress, nitrosation stress, and different classes of antimicrobials [41]. The AcrAB efflux pump has a multidrug-resistant effect on many bacteria. Previous studies revealed that, in *E. coli*, AcrAB-TolC efflux pumps can pump chloramphenicol, fluoroquinolones, tetracyclines, rifampicin, formidic acid, and β-lactam antibiotics out of the cell [42]. Other than that, the increase in the efflux system EmrKY-TolC and CusCFBA plays a crucial role in metformin-induced multiantibiotic resistance [43]. Based on these results, we hypothesized that CpxR could cause changes in resistance to multiple antibiotics by modulating efflux pump expression. We performed RT-qPCR to test this hypothesis. The results of RT-qPCR showed that CpxA/CpxR negatively influences transcription of the *emrKY* gene in APEC40, indicating that the Cpx signaling pathway also responds to the efflux pump gene *emrKY*. CpxR was shown to bind to the promoter region of the regulator σE in *E. coli* [44]. Therefore, we continued to explore the control mechanism of CpxR on the EmrKY efflux pump. It was proved by EMSA that CpxR can directly bind to the promotor region of the *emrKY* gene. It could be concluded that CpxR regulates the transcription level of *emrKY* efflux pumps by directly binding to the promoter region, thus effecting the resistance of the strain. In other words, the cpx pathway is an important pathway for APEC to reduce antibiotic susceptibility and improve viability.

In summary, our results showed that the CpxRA two-component system can directly regulate the expression of the *emrKY* efflux pump gene by its promoter region in APEC to affect strain drug resistance. However, whether and how CpxRA has a regulatory effect on other efflux pumps, virulence genes, etc. in APEC need to be further studied. Taken together, these results provide additional insights into the *E. coli* stress response network that relies on the CpxRA system and is a potential target for solving the infection problem.

## 5. Conclusions

In conclusion, the CpxRA system plays a critical role in regulating the biofilm formation ability and antibiotic resistance of *Escherichia coli* APEC40. The deletion of the *cpxRA* TCS has a serious inhibitory effect on APEC biofilm formation, and the amount of biofilm formation is reduced by 1.58-fold compared with WT. Moreover, the results of this study indicate a potential mechanism by which CpxR negatively regulates the expression of efflux pump gene *emrKY*, thereby influencing the resistance of APEC to erythromycin and ofloxacin.

## Figures and Tables

**Figure 1 animals-13-00383-f001:**
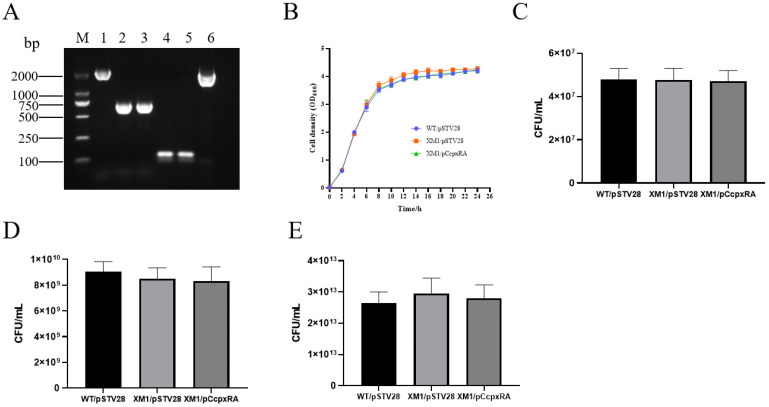
PCR identification and growth curve assays of WT, *cpxRA* mutant, and complementary strains. (**A**) Molecular identification of WT/pSTV28, XM1/pSTV28, XM1/pC*cpxRA* strains. M: 2000 bp marker, lanes 1–3 for the identification of *cpxRA* gene using primers Check-*cpxRA*-out-f/Check-*cpxRA*-out-r, lanes 4–6 for the identification of pSTV28 plasmid and complement plasmid pC*cpxRA* using primers M13-f/M13-r. (**B**) Growth curves of WT/pSTV28, XM1/pSTV28, XM1/pC*cpxRA* strains. (**C**–**E**) The number of colonies in culture medium at 2 h, 4 h, and 10 h, respectively.

**Figure 2 animals-13-00383-f002:**
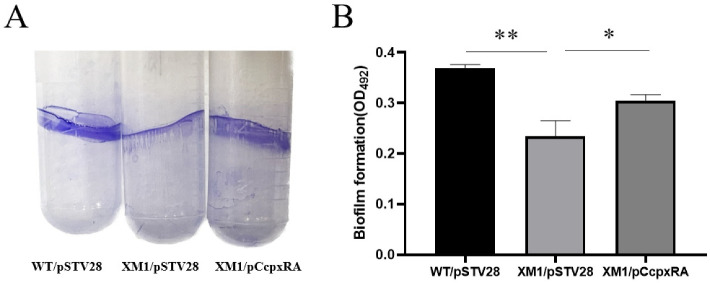
WT/pSTV28, XM1/pSTV28, and XM1/pC*cpxRA* biofilm formation ability detection. (**A**) Image of biofilm formation on the wall of a polystyrene tube. (**B**) Cells adhering to polystyrene tubes, 0.1% crystal violet staining, 33% glacial acetic acid dissolving, and measuring optical density at 492 nm. Error bars indicate SD; * represents *p* < 0.05, ** represents *p* < 0.01.

**Figure 3 animals-13-00383-f003:**
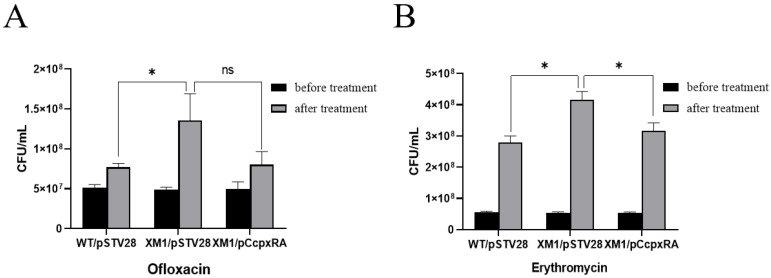
CFU assays of the WT/pSTV28, XM1/pSTV28, and XM1/pC*cpxRA* under two different antibiotic pressures: (**A**) ofloxacin, (**B**) erythromycin. Error bars indicate SD; * represents *p* < 0.05, ns represents no significant difference.

**Figure 4 animals-13-00383-f004:**
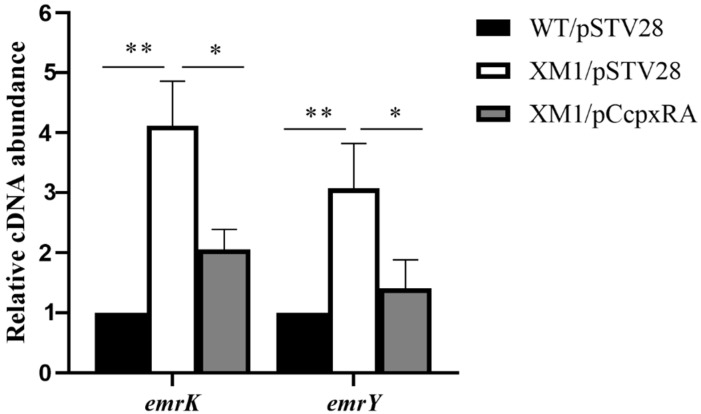
Transcript levels (cDNA abundance) of genes encoding efflux pumps were determined. The relative transcript levels of *emrK* and *emrY* in WT/pSTV28, XM1/pSTV28, and XM1/pC*cpxRA* were determined using RT-qPCR. Error bars indicate SD; * represents *p* < 0.05, ** represents *p* < 0.01.

**Figure 5 animals-13-00383-f005:**
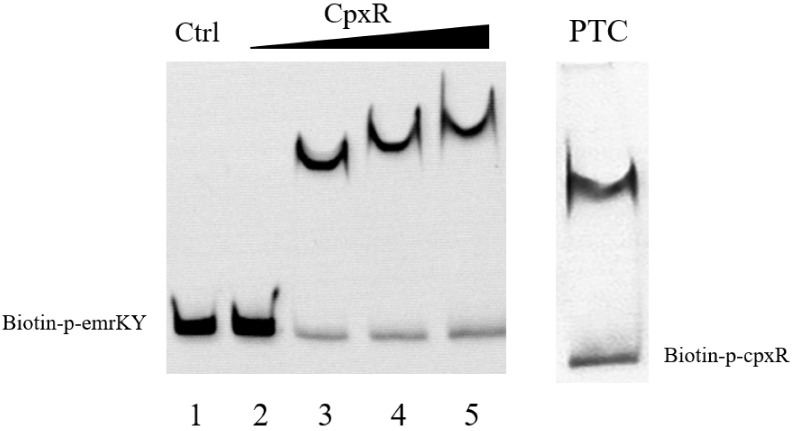
The binding ability of CpxR to the *emrKY* promoter was determined by EMSA. Lanes 1–5: the CpxR concentrations in order of 4, 0, 1, 2, and 4 μM, the amounts of biotin-labeled probes in all lanes were 200 fM, with an additional 2 pM of unlabeled probe added to lane 1 as a competition control (Ctrl). The positive control (PTC) indicated that CpxR was able to bind to its own promoter.

**Table 1 animals-13-00383-t001:** Strains and plasmids used in this study.

Strains or Plasmids	Description	Reference or Source
Strains		
*E. coli*		
DH5α	Clone host strain	Invitrogen
BL21(DE3)	Expression strain	Invitrogen
WT	Avian pathogenic *E. coli* (APEC) 40, wild-type	Laboratory stock
XM1	APECX40 *cpxAR-*deletion mutant	This study
WT/pSTV28	WT with the empty vector pSTV28, Cm^r^	This study
XM1/pSTV28	XM with the empty vector pSTV28, Cm^r^	This study
XM1/pC*cpxRA*	XM with the complement plasmid pC*cpxAR*, Cm^r^	This study
Plasmids		
pKD3	*cat* gene, template plasmid, Cm^r^ Amp^r^	[29]
pKD46	Expresses *λ* Red recombinase Exo, Bet, and Gam, temperature sensitive, Amp^r^	[29]
pCP20	*FLP*^+^*λc*I857^+^ *λ*_pR_Rep(Ts), temperature sensitive, Cm^r^ Amp^r^	[29]
pSTV28	Low copy number cloning vector, Cm^r^	Takara
pC*cpxRA*	pSTV28 with *cpxAR* gene, Cm^r^	This study
pET28a(+)	Expression vector, Kan^r^	Novagen
pET-*cpxR*	pET28a(+) with *cpxR* gene, Kan^r^	This study

**Table 2 animals-13-00383-t002:** Oligonucleotide primers used in this study.

Primer Name	Oligonucleotide (5′-3′)
APEC40-*cpxRA*-f	CGAGATGGAAGGCTTCAACGTGATTGTTGCCCACGATGGGTGTAGGCTGGAGCTGCTT
APEC40-*cpxRA*-r	CACCCAGCCACGATGCTGCTGAATGGCGGTTTCAACAATCTGAATATCCTCCTTAGTTC
CM-f	TGTAGGCTGGAGCTGCTT
CM-r	CATATGAATATCCTCCTTAGTTC
Check-*cpxRA*-in-f	TTTGACCGCCCGCTATTA
Check-*cpxRA*-in-r	CTTGCTTTCACCGCTACGAC
Check-*cpxRA*-out-f	ACTGACTGCCAGCGTTGA
Check-*cpxRA*-out-r	TGCCGGGTTATCGAAAAG
pET-*cpxR*-f	ACTTTAAGAAGGAGATATACCATGAATAAAATCCTGTTA
pET-*cpxR*-r	GTGGTGGTGGTGGTGCTCGAGTGAAGCAGAAACCATCAGAT
pC*cpxRA*-kpnⅠ-f	GCGGGTACCTGACGGCAGCGGTAACTA
pC*cpxRA*-hindⅢ-r	GCGAAGCTTAACTCCGCTTATACAGCG
M13-f	TGTAAAACGACGGCCAGT
M13-r	CAGGAAACAGCTATGACC
T7-f	TAATACGACTCACTATAGGG
T7-r	TGCTAGTTATTGCTCAGCGG
rt-16s-f	TTTGAGTTCCCGGCC
rt-16s-r	CGGCCGCAAGGTTAA
rt-*emrY*-f	CGATATTGGGCGGTTATA
rt-*emrY*-r	GGGTCAGTCCTGGTAGATT
rt-*emrK*-f	TGCCACTATCGCACTCAA
rt-*emrK*-r	GGTATGCTCCAGCGTTTC
p-*emrKY*-biotin-f	TCTCCCTTCTCCCTGTAGTAA
p-*emrKY*-r	TATTATCTCTCATTTCTCATAGATG

* The underlined sequences refer to the restriction endonuclease recognition sites.

**Table 3 animals-13-00383-t003:** Susceptibility of *Escherichia coli* strains to various antibiotics.

Antibiotics	MIC (μg/mL) of Three *E. coli* Strains		
	WT/pSTV28	XM1/pSTV28	XM1/pCcpxRA
Erythromycin	15	15	15
Ofloxacin	0.25	0.25	0.25

Abbreviation: MIC, minimal inhibitory concentration.

## Data Availability

The data presented in this study are available on request from the corresponding author.

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
