# Peer review of "The Two-Component System CpxRA Affects Antibiotic Susceptibility and Biofilm Formation in Avian Pathogenic Escherichia coli"

_animals, 2023, doi:10.3390/ani13030383_

Round 1
Reviewer 1 Report
Major comments.
Introduction section.
- Lines 53-56. It is lack the antimicrobial modification as a possible mechanism of antimicrobial resistance.
Results section.
- Line 228-230: The growth rate parameter is better for comparisons, it is suggested to include this value. At line 230 is mentioned the growth rate but Figure 1B showed the growth curve.
- Lines 254-255: The MICs values (or x-folds of difference) should be included, even more considering that authors mention the possible action of CpxRA over antibiotic susceptibility.
- Lines 259-261: erythromycin is a bacteriostatic antibiotic therefore the use of survival rate terminology is not correct.
- Figure 3 and lines 259-263: The survival rate values or microbial concentration defined by authors were relative to the total number of CFU of WT/pSTV28 (100%). A modification of 1.76-fold or 1.48-fold in this parameter could not be enough to make a conclusion. In order to evaluate the effect of some treatment against the bacterial growth, you should compare the initial inoculum (using CFU/ml or logCFU parameters) of each strain against the inoculum of each one after the treatment (3h antibiotic exposition).
- Lines 262-263: A higher microbial concentration relative value in the XM1/pSTV28 strain (cpxRA deleted) is not necessarily correlated with a reduced susceptibility to ofloxacin and/or erythromycin. If the XM1/pSTV28 strain is more susceptible to these drugs a lower MIC value is expected, therefore and for that reason, the MIC values should be included.
- Lines 275-276: The MIC value is mandatory to suggest that cpxRA gene influence over the antibiotic susceptibility.
- Lines 362-363: The conclusion “CpxR negatively regulates the drug resistance to ofloxacin and erythromycin…” could not be inferred from the showed results.
Minor comments.
- Check italics when write E. coli or Salmonella.
M&M section.
- Line 151: change CV for "crystal violet (CV)".
- Lines 163-164: It is ofloxacin/erythromycin concentration at the stock solution?
Results section.
- Line 231: “The results showed that the growth rate of WT/pSTV28, XM1/pSTV28 and XM1/pCcpxRA strains in LB medium containing 15 μg/mL chloramphenicol was like, indicating that the absence of cpxRA did not affect the growth of WT strain (Figure 1B).” The growth rate (if it is included) could “similar” instead “like”.
- Lines 241: change “straining” for “staining”
- Figure 3. Include the meaning of “ns” abbreviator.
Author Response
- Lines 53-56. It is lack the antimicrobial modification as a possible mechanism of antimicrobial resistance.
Reply: Thanks for your comments. The references [1,2] included antibacterial modification as a possible mechanism of antimicrobial resistance, part of which we listed. We have added this possible resistance mechanism in the revised manuscript. (Lines 54)
References
- Munita, J.M.; Arias, C.A. Mechanisms of Antibiotic Resistance. Microbiol Spectr 2016, 4, doi:10.1128/microbiolspec.VMBF-0016-2015.
- Camp, J.; Schuster, S.; Vavra, M.; Schweigger, T.; Rossen, J.W.A.; Reuter, S.; Kern, W.V. Limited Multidrug Resistance Efflux Pump Overexpression among Multidrug-Resistant Escherichia coli Strains of ST131. Antimicrob Agents Chemother 2021, 65, doi:10.1128/AAC.01735-20.
- Line 228-230: The growth rate parameter is better for comparisons, it is suggested to include this value. At line 230 is mentioned the growth rate but Figure 1B showed the growth curve.
Reply: We agree with your suggestion very much. Other reviewers also pointed out the imperfection of the growth curve, so we added the CFU counting to improve the experiment. (Lines 229 and Figure 1)
- Lines 254-255: The MICs values (or x-folds of difference) should be included, even more considering that authors mention the possible action of CpxRA over antibiotic susceptibility.
Reply: Thank you for your suggestion. According to your suggestion, we have added the MIC values in the revised manuscript. (Lines 258)
- Lines 259-261: erythromycin is a bacteriostatic antibiotic therefore the use of survival rate terminology is not correct.
Reply: We apologized for the inaccurate description. Erythromycin can bind to the 50S ribosome subunit of bacteria, and inhibit bacterial protein synthesis, thus inhibiting bacterial growth. We modified the expression in the revised manuscript and used CFU count to indicate the sensitivity of the strain to antibiotics. (Lines 264-267)
- Figure 3 and lines 259-263: The survival rate values or microbial concentration defined by authors were relative to the total number of CFU of WT/pSTV28 (100%). A modification of 1.76-fold or 1.48-fold in this parameter could not be enough to make a conclusion. In order to evaluate the effect of some treatment against the bacterial growth, you should compare the initial inoculum (using CFU/ml or logCFU parameters) of each strain against the inoculum of each one after the treatment (3h antibiotic exposition).
Reply: We are quite in favor of your suggestion. As you suggested, we used CFU counting to compare the effects of antibiotics on bacterial growth. (Lines 264-267)
- Lines 262-263: A higher microbial concentration relative value in the XM1/pSTV28 strain (cpxRA deleted) is not necessarily correlated with a reduced susceptibility to ofloxacin and/or erythromycin. If the XM1/pSTV28 strain is more susceptible to these drugs a lower MIC value is expected, therefore and for that reason, the MIC values should be included.
Reply: Thanks for your comments. We have shown MIC values in the manuscript. (Lines 258 and Table 3)
- Lines 275-276: The MIC value is mandatory to suggest that cpxRA gene influence over the antibiotic susceptibility.
Reply: Thanks for your comments. We have shown MIC values in the revised manuscript. (Lines 258)
- Lines 362-363: The conclusion “CpxR negatively regulates the drug resistance to ofloxacin and erythromycin…” could not be inferred from the showed results.
Reply: We apologize for the inaccurate expression. We have changed this sentence to "The results of this study indicate a potential mechanism by which cpxR negatively regulates the expression of efflux pump gene emrKY, thereby influencing the resistance of APEC to erythromycin and ofloxacin." in the manuscript. (Lines 368-371)
- Check italics when write E. coli or Salmonella.
Reply: Thanks for your comments. We checked the full text and corrected it.
- Line 151: change CV for "crystal violet (CV)".
Reply: Thanks for your comments. We have revised it in the manuscript. (Lines 152)
- Lines 163-164: It is ofloxacin/erythromycin concentration at the stock solution?
Reply: Thanks for your comments. The concentration of erythromycin and ofloxacin is 30 mg/mL and 20 mg/mL respectively. We have shown the stock concentration of antibiotics in the revised manuscript. (Lines 164-165)
- Line 231: “The results showed that the growth rate of WT/pSTV28, XM1/pSTV28 and XM1/pCcpxRA strains in LB medium containing 15 μg/mL chloramphenicol was like, indicating that the absence of cpxRA did not affect the growth of WT strain (Figure 1B).” The growth rate (if it is included) could “similar” instead “like”.
Reply: Thanks for your comments. We have modified it according to your suggestion. (Lines 232)
- Lines 241: change “straining” for “staining”
Reply: Thanks for your comments. We have revised it in the revised manuscript. (Lines 245)
- Figure 3. Include the meaning of “ns” abbreviator.
Reply: Thanks for your comments. We have added the meaning of “ns” abbreviation to the figure legends. (Lines 274)
Reviewer 2 Report
In their paper, Ma et al. describe the role of the two-component system CprRA in antibiotic susceptibility and biofilm formation in APEC. The paper is very well written. However, to be published in Animals, the paper should be improved.
Major comments:
1- In their introduction, the authors claimed that "this study suggested that CpxRA is a potential drug target for the prevention or treatment of colibacillosis." (lines 85-86). To me, with their results, it's too ambitious to put this sentence ahead. We don't even know yet if CpxRA is expressed upon infection...
2- Figure 1B: the authors measured the OD600 and claim that there is no growth differences between the WT strain and the mutant. The OD600 is not accurate enough to demonstrate that the growth is similar. Indeed, the shape of the bacteria could be modified by the deletion of the 2 genes of cpxRA operon, involved in the envelope stress response, and one can imagine that this could affect the shape and/or growth of E. coli. Then, to determine that CpxRA doesn't affect growth, the authors should complete their OD600 measurement with a CFU count.
3- In the conclusion, the authors claimed that "Moreover, CpxR negatively regulates the drug resistance to ofloxacin and erythromycin by directly binding to the promoter region of the efflux pump gene emrKY". (lines 362-364). They can't affirm that since they didn't show directly by a mutant in cpxRA AND emrKY that they abolished the resistance of their APEC. They can only say that the results SUGGEST.
Minor suggestions:
1- line 73 "which is consist of a sensor" should be rephrased as "which consist..."
2- line 77 "resistance to APEC regulated by BasSR TCS". What is the BasSR TCS? Did the author want to say CpxRA?
3- line 112 "the cpxRA gene deletion mutant". It is confusing. Did the authors delete only one gene of cpxRA operon or both geneS? This is confusing all over the manuscript and should be clarified everywhere else in the manuscript.
4- line 151 "with 0,1% (W:v) CV" what is CV? Crystal violet?
5- line 275 "These data suggested that cpxRA gene influences the antibiotic sensitivity". Here, the authors should be more precise. Instead of "influences", which can be related to activation OR inhibition, they should mention that the operon decreases the antibiotic sensitivity of APEC.
6- line 284 "EMSA experiments were performed to verify this speculation". EMSA experiments reinforce this hypothesis, but since it's in vitro assay (different from ChIP assay) and the conditions in the tube could bring a bias to the binding, the authors could not "verify this speculation" by EMSA. This must be rephrased with something like "reinforce the hypothesis".
7- line 332 "Salmonella Typhimurium". Salmonella is a genus and must be written in italic "Salmonella Typhiumurium"
Author Response
- In their introduction, the authors claimed that "this study suggested that CpxRA is a potential drug target for the prevention or treatment of colibacillosis." (lines 85-86). To me, with their results, it's too ambitious to put this sentence ahead. We don't even know yet if CpxRA is expressed upon infection...
Reply: We apologized for the imperfect description. We have changed the language this sentence to “This study helps to understand the regulatory mechanism of APEC under antibiotic pressure, and provides a new experimental basis for the treatment of E.coli infection.” in the revised manuscript. (Lines 85-86)
- Figure 1B: the authors measured the OD600 and claim that there is no growth differences between the WT strain and the mutant. The OD600 is not accurate enough to demonstrate that the growth is similar. Indeed, the shape of the bacteria could be modified by the deletion of the 2 genes of cpxRA operon, involved in the envelope stress response, and one can imagine that this could affect the shape and/or growth of E. coli. Then, to determine that CpxRA doesn't affect growth, the authors should complete their OD600 measurement with a CFU count.
Reply: We are quite in favor of your suggestion. As you suggested, we added CFU counting experiments to prove that the growth was similar. (Lines 141-142 and Figure 1)
- In the conclusion, the authors claimed that "Moreover, CpxR negatively regulates the drug resistance to ofloxacin and erythromycin by directly binding to the promoter region of the efflux pump gene emrKY". (lines 362-364). They can't affirm that since they didn't show directly by a mutant in cpxRA AND emrKY that they abolished the resistance of their APEC. They can only say that the results suggest.
Reply: Thanks for your comments. We have changed this sentence to "The results of this study indicate a potential mechanism by which cpxR negatively regulates the expression of efflux pump gene emrKY, thereby influencing the resistance of APEC to erythromycin and ofloxacin." in the revised manuscript. (Lines 368-371)
- line 73 "which is consist of a sensor" should be rephrased as "which consist..."
Reply: Thanks for your comments. We revised it in the revised manuscript. (Lines 73)
- line 77 "resistance to APEC regulated by BasSR TCS". What is the BasSR TCS? Did the author want to say CpxRA?
Reply: We apologize for the mistake we made. We have modified this sentence and modified BasSR to CpxRA in the manuscript. (Lines 77)
- line 112 "the cpxRA gene deletion mutant". It is confusing. Did the authors delete only one gene of cpxRA operon or both geneS? This is confusing all over the manuscript and should be clarified everywhere else in the manuscript.
Reply: We apologize for the inaccurate description. We deleted all the cpxRA operon genes, which we have clarified in the appropriate places in the revised manuscript. (Lines 112)
- line 151 "with 0,1% (W:v) CV" what is CV? Crystal violet?
Reply: We apologized for the inaccurate description. We have changed CV for crystal violet (CV) in the manuscript. (Lines 152)
- line 275 "These data suggested that cpxRA gene influences the antibiotic sensitivity". Here, the authors should be more precise. Instead of "influences", which can be related to activation OR inhibition, they should mention that the operon decreases the antibiotic sensitivity of APEC.
Reply: Thanks for your comments. According to your suggestions, we have made a precise description in the revised manuscript. (Lines 281)
- line 284 "EMSA experiments were performed to verify this speculation". EMSA experiments reinforce this hypothesis, but since it's in vitro assay (different from ChIP assay) and the conditions in the tube could bring a bias to the binding, the authors could not "verify this speculation" by EMSA. This must be rephrased with something like "reinforce the hypothesis".
Reply: We are quite in favor of your proposal. EMSA assays is an in vitro experiment to prove the mechanism of interaction between transcription regulators and target genes, and this method is used in most articles [1-3]. Indeed, ChIP sequencing can further prove this hypothesis, which will be performed in our future work. According to your suggestion, we have corrected this expression. -(Lines 289-290)
Reference:
- Gerken H, Vuong P, Soparkar K, Misra R. Roles of the EnvZ/OmpR Two-Component System and Porins in Iron Acquisition in Escherichia coli. mBio. 2020 Jun 23;11(3):e01192-20. doi: 10.1128/mBio.01192-20.
- Wang H, Dong L, Wu W, Hu H, Zhang LH, Liao L. RdmA Is a Key Regulator in Autoinduction of DSF Quorum Quenching in Pseudomonas nitroreducens HS-18. mBio. 2022 Dec 20:e0301022. doi: 10.1128/mbio.03010-22.
- Mlynek KD, Sause WE, Moormeier DE, Sadykov MR, Hill KR, Torres VJ, Bayles KW, Brinsmade SR. Nutritional Regulation of the Sae Two-Component System by CodY in Staphylococcus aureus. J Bacteriol. 2018 Mar 26;200(8):e00012-18. doi: 10.1128/JB.00012-18.
- line 332 "Salmonella Typhimurium". Salmonella is a genus and must be written in italic "Salmonella Typhiumurium"
Reply: Thanks for your comments. We have revised it in the revised manuscript. (Lines 338)
Round 2
Reviewer 1 Report
The authors carried out the suggestions and the manuscript was substantially improved.